# A scholarly network of AI research with an information science focus: Global North and Global South perspectives

**Kai-Yu Tang**[1]*, **Chun-Hua Hsiao**[2], **Gwo-Jen Hwang**[3]

**1** Department of International Business, Ming Chuan University, Taipei, Taiwan, **2** School of Business, Kainan University, Taoyuan, Taiwan, **3** Graduate Institute of Digital Learning and Education, National Taiwan University of Science and Technology, Taipei, Taiwan

* ky.nctu@gmail.com

**Data Availability Statement:** The dataset has been uploaded to Harvard Dataverse: https://doi.org/10.7910/DVN/KACCSU.

**Funding:** This study is partially supported by the Ministry of Science and Technology, Taiwan, under

## Abstract

This paper primarily aims to provide a citation-based method for exploring the scholarly network of artificial intelligence (AI)-related research in the information science (IS) domain, especially from Global North (GN) and Global South (GS) perspectives. Three research objectives were addressed, namely (1) the publication patterns in the field, (2) the most influential articles and researched keywords in the field, and (3) the visualization of the scholarly network between GN and GS researchers between the years 2010 and 2020. On the basis of the PRISMA statement, longitudinal research data were retrieved from the Web of Science and analyzed. Thirty-two AI-related keywords were used to retrieve relevant quality articles. Finally, 149 articles accompanying the follow-up 8838 citing articles were identified as eligible sources. A co-citation network analysis was adopted to scientifically visualize the intellectual structure of AI research in GN and GS networks. The results revealed that the United States, Australia, and the United Kingdom are the most productive GN countries; by contrast, China and India are the most productive GS countries. Next, the 10 most frequently co-cited AI research articles in the IS domain were identified. Third, the scholarly networks of AI research in the GN and GS areas were visualized. Between 2010 and 2015, GN researchers in the IS domain focused on applied research involving intelligent systems (e.g., decision support systems); between 2016 and 2020, GS researchers focused on big data applications (e.g., geospatial big data research). Both GN and GS researchers focused on technology adoption research (e.g., AI-related products and services) throughout the investigated period. Overall, this paper reveals the intellectual structure of the scholarly network on AI research and several applications in the IS literature. The findings provide research-based evidence for expanding global AI research.

## Introduction

The development of the information science (IS) domain through the application of artificial intelligence (AI) has had profound effects in numerous industries (e.g., finance, health care,

contract numbers MOST 110-2511-H-130-001 (Kai-Yu Tang), MOST 109-2410-H-424-003 (Chun-Hua Hsiao), and MOST 109-2511-H-011-002-MY3 (Gwo-Jen Hwang). The funders had no role in study design, data collection and analysis, decision to publish, or preparation of the manuscript.

**Competing interests:** The authors have declared that no competing interests exist.

manufacturing, retail, supply chain, logistics, and utilities). According to the literature, AI is defined as "a system that perceives its environment and takes actions to maximize its ability to achieve its goals" [1, 2]. Researchers have also suggested that the development of supercomputing power has made AI the crucial emerging theme in the IS research domain [3]. For example, Nishant et al. [4] highlighted five challenges to sustainability in AI research (e.g., overreliance on machine learning for historical data, unpredictable human responses to AI interventions, and difficulties in measuring the effects of AI interventions). Duan et al. [3] identified the challenges associated with the use and effect of AI on decision-making in IS research (e.g., theoretical development, technology–human interaction, and AI implementation). Recently, Dwivedi et al. [5] presented a holistic view of AI research from the perspective of information science; on the basis of the insights of experts, they categorized existing research into major themes (e.g., decision making, application, and data), domains (e.g., governance, science, and technology), and key challenges.

In addition, IS researchers have highlighted the key role of big data research in the development of AI technology. For example, Surbakti et al. [6] identified factors that influence the effective use of big data to provide guidance for organizations; they identified seven themes that were divided into three groups, namely motivation (perceived organizational benefit), operation (process management, human aspects, systems, and organizational aspects), and supporting mechanisms (e.g., data quality, privacy, and governance). On the basis of an analysis of 60 published articles, Lugmayr et al. [7] proposed a five-trait framework for cognitive data; the five traits are sociotechnical systems, data space, data richness, decision-making, and visualization. Baig et al. [8] identified theoretical models and factors that influence big data adoption; they discovered that the diffusion of innovations and technology–organization–environment are the most popular theoretical models and identified 42 antecedents in technology, organization, environment, and innovation.

Scholars have also conducted bibliometric analyses to examine the knowledge structure of AI-related applications in various research areas. For example, de Sousa et al. [9] conducted a systematic literature review to identify research topics and trends relating to AI in the public sector. Chen et al. [10, 11] performed a bibliometric analysis to investigate AI-related applications in the medical field, such as an AI-enhanced electroencephalogram method for conducting human brain research. Other educators have combined bibliometric and content analysis to explore mainstream AI-assisted education research [12, 13]. Review studies have also reported that comparing different research periods can help researchers to identify potential trends in a research field. For instance, de Sousa et al. [9] reviewed AI studies that were conducted in the public sector between 2000 to 2018 and compared each 3-year period within this timeframe to identify changes in publication patterns. Chen et al. [10] compared the AI-assisted brain research published between the years 2010 and 2014 and between the years 2015 and 2018. Hwang and Tu [13] conducted a three-stage (1996–2010, 2011–2015, 2016–2020) survey to analyze AI trends in mathematics education. The aforementioned studies reviewed the effects of AI-related technologies on mathematical learning and related research trends. Past research has provided insights about the development of AI-related literature. However, to the best of our knowledge, few studies have explored the intellectual structure of scholarly networks in AI-related literature by applying a citation network method and conducting a geographic region–based comparison, especially from the perspective of the Global North and Global South.

Co-citation network analysis is regarded as a well-established analytic method for exploring the cited-and-citing relationships between two focal articles [14]. Researchers have adopted the method to analyze the intellectual structure of various themes, including e-book-supported learning [15] and virtual reality for education [16]. Garfield [17] suggested that citations can

be treated as an index of science that provides evidence for measuring the effect of published research articles and directions on how to extend published research. The more frequently two focal articles are cited together, the higher the bibliometric similarity between the two articles is. Small [14] defined the occurrence of co-citation as the citation of two or more previously published focal articles together by a subsequent peer-reviewed article, which illuminates how researchers evaluate previous publications in a field. Therefore, researchers have proposed that co-citation analysis is a scientific measure that represents the intellectual structure of academic problem spaces [18]. Moreover, we contend that a visualized co-citation network provides a structural understanding of the IS scholarly network on AI research. Through an exploration of highly co-cited articles, mainstream research patterns (e.g., keywords and research foci) can be scientifically identified.

For global development research, North–South divisions are often based on the political and socioeconomic dimensions of the literature (e.g., Martin, 1988). However, on the basis of the high growth rate of a "scientific culture" in several countries [19], such as China and India, several researchers have suggested that scientific development can be explored from the perspective of the Global North (GN) and Global South (GS) [19]. This typology meets the primary purpose of the present study, which is to explore global AI-related research developments. Therefore, in accordance with the research conducted by Confraria et al. [19], we defined the GN to include North America, Western Europe, and the developed regions of East Asia and defined the GS to include Africa, Latin America, developing regions of Asia, and the Middle East.

In the present study, a co-citation network analysis was conducted to compare the GN and GS with respect to scholarly networks of AI research in the IS domain during the period from 2010 to 2020. Furthermore, to allow for research trend comparisons with the findings of other studies [10–13], the research period for this study was divided into two intervals (i.e., 2010–2015 and 2016–2020). Overall, the present study provides a fresh perspective on the global research dynamics of the scholarly network of AI research in the IS field. The three research questions are specified as follows:

- 1) Investigation of the publication patterns of AI research in the IS domain (RQ1): From the bibliometric perspective, what are the publication patterns and the most productive countries in the investigated field between 2010 and 2015 and between 2016 and 2020?

- 2) Identification of the most frequently cross-referenced AI research articles in the IS domain (RQ2): On the basis of follow-up citations, the present study performed a co-citation analysis to determine the core articles of IS field. The main AI research foci of GN and GS researchers were also analyzed.

- 3) Visualization of the global scholarly network on AI research in the IS domain (RQ3): From the social networking perspective, what is the intellectual structure of the scholarly IS network on AI research? Through the application of social network analysis and the clustering technique, the citation-based mapping of scholarly networks of AI research in the IS domain was achieved.

## Methods

### Ethics statement

In the present study, a retrospective bibliometric analysis that focused on analyzing published articles was conducted. No clinical trials were conducted in the present study.

## Research data: Inclusion and categorization

The process of including research data is crucial for a literature analysis. In accordance with the Preferred Reporting Items for Systematic Reviews and Meta-Analyses (PRISMA) statement [20], a four-step procedure for data inclusion was applied in the present study to systematically identify, screen, select, and include the most relevant AI-related research articles in the IS domain (Fig 1).

First, on the basis of review studies of the related literature [3–5], 32 AI-related keywords (see S1 Appendix) were used to conduct an opening search to retrieve all relevant articles. In line with the main research interest of the present study, a keyword search of the category of "library and information science" was then conducted on the Web of Science (WoS) database. Sixty-three out of the 87 journals (IS focused) in the WoS relevant to the aforementioned category were included for the search. Second, a series of screening processes were performed. For example, we used a topic search to systematically filter out irrelevant articles that did not contain any search keywords in their article titles, keywords, and abstracts. We also used Boolean functions to exclude redundant items from our searches and set a citation requirement to maintain both the quality of our research and that of the subsequent citation-based analysis. During the data search, 485 out of 677 articles with less-than-average citation frequency (3.9 times per year) were excluded. Third, on the basis of the aforementioned systematic check, 149 articles and a corresponding 8838 follow-up citing articles were identified as eligible sources; they represented the intellectual structure of the field for the subsequent analysis. Only journal citation records from the Science Citation Index Expanded (i.e., SCI-E) and Social Sciences

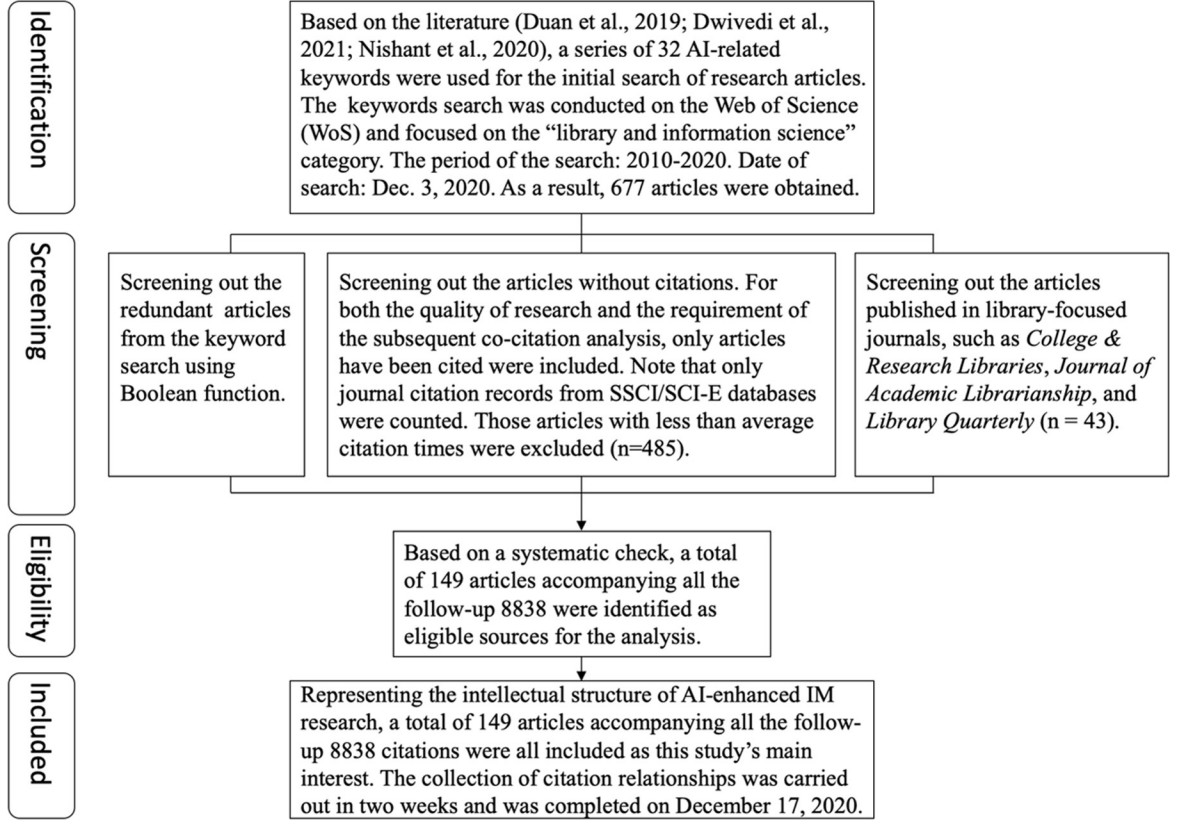

**Fig 1. PRISMA diagram for inclusion of research data.**

Citation Index (i.e., SSCI) databases were counted to maintain the quality of research. Finally, the complete data on cited-and-citing relationships (i.e., 149 core articles and the corresponding 8838 follow-up citing articles) were collected document by document. The search was conducted on December 3, 2020, and the collection of data on citation relationships was performed over 2 weeks and completed on December 17, 2020.

Furthermore, we retrieved bibliometric data on the affiliations of first authors as provided in the included articles. In accordance with other author co-citation studies [18], the affiliations of first authors were used as the basis for determining the affiliated country for each article. To present an overview of global research productivity in the investigated field, the most productive countries were analyzed. The results indicated that all the first authors of the 149 articles were from 31 countries. A further categorization of the GN and GS was conducted based on the definition proposed by Confraria et al. [19].

## Co-citation network analysis

Based on a systematic screening process, the collection of 149 high-quality papers stands for an invisible research community of IS scholarly network on AI research. According to the WoS, 149 articles were cited 8838 times (date of search: December 23, 2020). The lens from the great number of researchers that joint citing the 149 articles (i.e., 8838 follow-up studies) may provide collective wisdom of self-reflection and the field's view of itself to review the area. Among the available bibliometric methods, the recent literature has suggested that co-citation analysis provides citation-based evidence for measuring the relationships among scientific publications [15, 16]. A co-citation network analysis was thus performed to examine the IS scholarly network on AI research and determine how the aforementioned 149 core articles are related to each other from the perspectives of all 8838 citing references.

The co-citation network analysis performed in the present study integrated the advantages of document co-citation analysis (DCA) and social network analysis (SNA). In the context of the present study, DCA was used to calculate the cross-referenced relationships among selected articles, and SNA was used to visualize a networking structure on the basis of a co-citation matrix to reveal the underlying structure of an IS scholarly network on AI research. The co-citation network analysis was conducted in the two steps as follows. First, a DCA was performed as the initial attempt to examine the co-citation relationships among the 149 core articles. Originating from bibliometrics, DCA is a quantitative analytic method for identifying the most frequently joint-cited research chain in a field. According to Small [14], when two earlier documents (cited articles) are jointly referenced by a follow-up document (citing articles), an instance of co-citation has occurred. In the present study, a symmetric raw co-citation matrix was formed to represent the co-citation relationships of 149 core AI-related research articles in the investigated field. In addition, the core articles in the matrix were then paired and matched if they were co-cited by citing articles. The higher the co-citation value of a given pair of core articles was, the greater was the bibliometric similarity of these two core articles.

Second, an SNA was used to visualize the co-citation network of an IS scholarly network on AI research, thereby establishing an overall social networking structure of the invisible colleagues in the field. Wasserman and Faust [21] indicated that the key value of SNA is as a novel visualization method for mapping the most prominent documents in a network. Moreover, Fahmy and Young [22] suggested that analyzing co-citation networks provides researchers with opportunities to access other researchers who were previously unknown to them; this is because the interactions that occur in a bibliometric context resemble a network of invisible colleagues. Hence, in the present study, a raw co-citation matrix was used as the data input for visualizing an AI research network. The bibliometric mapping method has been used to

represent the "invisible colleagues" in various fields, including health literacy research [23] and international workplace violence research [24], and for the representation of an open knowledge network [25]. Researchers also proposed the use of SNA to detect social information in scientific document databases to establish an intelligent recommendation system for researchers [26]. In the same vein, we referenced the aforementioned studies and adopted co-citation network analysis to explore the research patterns of an IS scholarly network with an AI research focus. In addition, the scholarly networks of GN and GS researchers were visualized, thereby providing a contemporary IS research frontier for the investigated field.

## Results

### Publication patterns of AI research in IS domain (RQ1)

In terms of the research productivity of global researchers in the investigated field, the most productive countries from the GN and GS were analyzed. The bibliometric results presented in Table 1 reveal that researchers from the United States were the main contributors (48 articles) to the investigated field. The United States is also the most productive country in the GN area for the two investigated periods (i.e., 2010–2015 and 2016–2020). The other high-ranked GN countries included Australia and the United Kingdom, which each published seven articles and were co-ranked at fourth place, and Canada, South Korea, Italy, Taiwan, and the Netherlands, which each published five articles and were co-ranked at sixth place. The most frequently cited researchers in the field were from the United States and Canada; the total number of citations of published articles from the United States and Canada was 4094 and 1167, respectively. In terms of productivity of countries in the GS area, the results revealed that

**Table 1. Top 20 most productive countries in terms of AI research in information science domain.**

| # | Country | Total Articles | Articles (2010–2015) | Articles (2016–2020) | Total Citations | GN | GS |
|---|---|---|---|---|---|---|---|
| 1 | USA | 48 | 28 | 20 | 4094 | V | |
| 2 | China | 20 | 4 | 16 | 610 | | V |
| 3 | India | 9 | 1 | 8 | 260 | | V |
| 4 | Australia | 7 | 0 | 7 | 272 | V | |
| 4 | UK | 7 | 1 | 6 | 208 | V | |
| 6 | Canada | 5 | 1 | 4 | 1167 | V | |
| 6 | South Korea | 5 | 3 | 2 | 386 | V | |
| 6 | Italy | 5 | 0 | 5 | 181 | V | |
| 6 | Taiwan | 5 | 2 | 3 | 114 | V | |
| 6 | Netherlands | 5 | 2 | 3 | 104 | V | |
| 11 | Malaysia | 4 | 1 | 3 | 423 | | V |
| 12 | Denmark | 3 | 1 | 2 | 185 | V | |
| 12 | Switzerland | 3 | 0 | 3 | 60 | V | |
| 13 | Germany | 2 | 2 | 0 | 181 | V | |
| 13 | Spain | 2 | 0 | 2 | 118 | V | |
| 13 | Brazil | 2 | 0 | 2 | 94 | | V |
| 13 | Portugal | 2 | 0 | 2 | 64 | V | |
| 13 | France | 2 | 0 | 2 | 53 | V | |
| 13 | Norway | 2 | 0 | 2 | 33 | V | |
| 13 | Pakistan | 2 | 0 | 2 | 11 | | V |
| | Subtotal (top 20) | 140 | 46 | 94 | 8618 | | |
| | Subtotal (others) | 9 | 1 | 8 | 220 | | |
| | Total (31 countries) | 149 | 47 | 102 | 8838 | 111 | 38 |

China and India were the two most productive countries. Researchers from China published 20 articles with 610 citations, following by India with nine articles published and 260 citations. The other GS countries in the top 20 list were Malaysia, which published four articles and was ranked at 11th place, and Brazil and Pakistan, which each published two articles and were co-ranked at 13th place. Overall, the top 20 countries accounted for more than 94% of the published articles and 97% of the total citations in the field.

On the basis of the aforementioned results, several changes in publication patterns relating to AI research in the IS domain were identified. First, 47 articles were published in the earlier investigated period (2010–2015), and the number of published articles increased to 102 during the subsequent five years (2016–2020). This result revealed an increasing trend in terms of the number of AI-related articles published globally in recent years. Specifically, during the earlier investigated period, the 41 published articles (87%) were mainly written by GN researchers, and only six articles (13%) were published by GS researchers. In the subsequent period (2016–2020), GN researchers published 70 articles, representing 73% of the increase in the number of articles published in the investigated field. However, GS researchers published 32 articles in the subsequent period, which was equivalent to a considerable increase of 433% relative to the earlier investigated years (2010–2015). From the GN and GS perspective, the change in publication patterns indicates the increasing presence of GS researchers in AI research; this finding echoes that of Confraria et al. [19], who reported the high growth of "scientific culture" in several GS countries.

Concerning the trends among influential journals for AI research in the IS domain, the present study also analyzed and compared international journal publications in the investigated field. Overall, the results revealed that the top four journals published at least 10 articles relating to the investigated field, whereas the journals ranked between fifth and tenth place published 5.5 articles on average. Notably, the top four journals were the *International Journal of Information Management* (*IJIM*; n = 32,), *Journal of the American Medical Informatics Association* (*JAMIA*; n = 26), *Information & Management* (n = 14), and *Journal of Knowledge Management* (n = 10). Our results revealed that the articles published in the IJIM and JAMIA were cited approximately 4000 times.

On the basis of a comparison of the articles published by GN and GS researchers, we discovered that two-third of the articles (21 out of 32) published in the IJIM were mainly contributed by GN researchers. Most of JAMIA articles (25 out of 26) in the field was authored by GN researchers. Similarly, the articles published in the *MIS Quarterly* (n = 5) and the *MIS Quarterly Executive* (n = 5), which were co-ranked at seventh place, were mostly written by GN researchers. However, GS researchers outperformed GN researchers with respect to the publication of articles concerning the investigated field in the journals *Information Processing & Management* (4 out of 7), *International Journal of Geographical Information Science* (4 out of 5), and *Scientometrics* (5 out of 5). Overall, the top 10 journals had a total of 115 published articles (77%) on AI-related research in the IS domain, and these articles were co-cited 7655 times (87% of total citations). In terms of publication performance, the results of the bibliometric analysis suggested that the top 10 journals were the most influential IS journals on AI research.

## Top cross-referenced articles on AI research in IS domain (RQ2)

The aforementioned bibliometric analysis results provided a macrolevel overview of the development of AI research. This section discusses the results of the co-citation analysis conducted in the present study to identify the bibliometric relationships among 149 selected articles, on the basis of which a microlevel understanding of the most influential articles in the investigated field can be established. The co-citation analysis drew from the collective wisdom of numerous

**Table 2. Co-citation matrix of top 10 cross-referenced articles.**

| # | ID | 1 | 2 | 3 | 4 | 5 | 6 | 7 | 8 | 9 | 10 | Co-Citations |
|---|---|---|---|---|---|---|---|---|---|---|---|---|
| 1 | GN-ChenCS2012 [27] | 0 | | | | | | | | | | 912 |
| 2 | GN-GandomiH2015 [29] | 160 | 0 | | | | | | | | | 733 |
| 3 | GN-GuptaG2016 [34] | 55 | 32 | 0 | | | | | | | | 409 |
| 4 | GN-ChenPS2015 [30] | 50 | 30 | 27 | 0 | | | | | | | 382 |
| 5 | GN-KwonLS2014 [31] | 60 | 57 | 22 | 21 | 0 | | | | | | 357 |
| 6 | GN-ConstantiouK2015 [32] | 35 | 9 | 16 | 14 | 7 | 0 | | | | | 209 |
| 7 | GN-Raguseo2018 [35] | 14 | 29 | 7 | 7 | 12 | 1 | 0 | | | | 194 |
| 8 | GS-RehmanCBT2016 [28] | 18 | 25 | 7 | 13 | 8 | 1 | 9 | 0 | | | 196 |
| 9 | GN-GroverCLZ2018 [36] | 21 | 10 | 16 | 10 | 7 | 7 | 1 | 1 | 0 | | 181 |
| 10 | GN-LoebbeckeP2015 [33] | 33 | 9 | 8 | 12 | 4 | 13 | 1 | 1 | 2 | 0 | 166 |

citing scholars (i.e., 8838 citing references) and provided a scientific measurement model for counting co-citations to evaluate the relationships between focal article pairs [14]. In the present study, the 8838 references that cited the 149 core articles were treated as the field's view of itself and considered to reveal the most influential AI-related research in the IS domain. In the co-citation analysis, the information of all 149 core articles was tabulated to create an asymmetric co-citation matrix. Each cell in the matrix was counted by matching the frequency of co-citation with the 8838 references. Co-citations represented the level of bibliometric similarity between two focal articles of interest that were referenced by the citing researchers.

On the basis of the co-citation analysis, 8976 co-citations were counted. Overall, each top-10 article was co-cited with another of the 149 selected articles more than 100 times (between 166 and 912 times) by the follow-up citing articles. The total number of co-citations received by the top 10 co-cited articles was 3739, which accounted for approximately 42% of the total number of co-citations (n = 8976). Table 2 presents that most of the frequently referenced articles were mainly written by GN researchers; nine out of the top 10 articles were written by GN researchers (e.g., Chen et al. [27]), and only one was from the GS researchers (e.g., Rehman et al. [28]). In the present study, highly co-cited articles in the matrix were identified on the basis of GS/GN information, authorship, and year of publication. The two articles, for instance, were labeled as "GN-ChenCS2012", and "GS-RehmanCBT2016". From the perspective of follow-up researchers, the aforementioned finding regarding citations indicated that the top 10 articles were frequently referenced together by field researchers, and thus, they represented the central research in the investigated field.

Table 2 lists the highly co-cited articles on AI research in the IS domain, and it reveals that 60% (6 of 10) of such articles were published during the earlier investigated period (2010–2015); the 60% comprised Chen et al. (#1) [27], Gandomi and Haider (#2) [29], Chen et al. (#4) [30], Kwon et al. (#5) [31], Constantiou and Kallinikos (#6) [32], and Loebbecke and Picot (#10) [33]. The other 40% of the highly referenced articles were published in the recent five years (2016–2020); they comprised Gupta and George (#3) [34], Raguseo (#7) [35], Rehman et al. (#8) [28], and Grover et al. (#9) [36]. Of note, these highly referenced articles mainly focused on big data–related research, including big data analytics (n = 7; e.g., Chen et al. [27]), big data technologies [29], and big data capabilities [34].

For the most researched AI-related keywords, the author-defined keywords that appeared in the 149 articles included in the present study were collected and counted. Table 3 lists the top five keywords and provides information on the number of articles in which each keyword is used. An overall comparison of the frequency of keyword use during the two investigated periods (2010–2015 and 2016–2020) revealed that "big data" was the most studied topic (used

**Table 3. Most researched AI-related keywords among Global South and Global North scholars (2010–2020).**

| # | Year | 2010–2020 | 2010–2015 (P$_1$) | | | 2016–2020 (P$_2$) | | |
|---|------|-----------|-----|-----|-------------|-----|-----|-------------|
| | | Total | GS | GN | Sub-total$_1$ | GS | GN | Sub-total$_2$ |
| 1 | Big data / Big data analytics | 105 | 2 | 19 | 21 | 27 | 57 | 84 (↑) |
| 2 | Machine learning | 20 | 1 | 6 | 7 | 5 | 8 | 13(↑) |
| 3 | Text mining / Data mining | 19 | 2 | 10 | 12 | 1 | 6 | 7(↓) |
| 4 | Natural language processing | 14 | 0 | 8 | 8 | 1 | 5 | 6(↓) |
| 5 | Neural network | 7 | 1 | 0 | 1 | 1 | 5 | 6(↑) |
| | Total | 166* | 6 | 43 | 49 | 35 | 82 | 116 |
| | Increase (%) | | | | | 483% | 88% | 137% |

*Researchers may use multiple keywords to characterize the focal interest of their article. Therefore, the total number of times the top five keywords may exceed the number of analyzed articles (n = 149); however, the maximum number of times each keyword was used was 149 in the present analysis.

by 105 articles), especially during the later five years (increase from 21 articles between 2010 and 2015 to 84 articles between 2016 and 2020). In addition to "big data," the other top five keywords were "machine learning (20)," "text/data mining (19)," "natural language processing (14)," and "neural network (7)." Although the keywords "machine learning" (increase from 7 to 13) and "neural network" (increase from 1 to 6) exhibited increasing trends in terms of frequency of use, the keywords "natural language processing" (decrease from 12 to 7) and "neural network" (decrease from eight to six) exhibited slight downward trends from the first to the second investigated periods.

For the top five most researched topics pertaining to AI, both GN and GS researchers collectively published 49 articles in the earlier investigated period (2010–2015), and this number increased to 116 in the second investigated period (2016–2020), exhibiting a 137% overall growth rate. From a global development perspective, the GN researchers achieved a low growth rate (88%) but a considerable increase in the number of published articles between the two periods (from 43 to 82 articles). By contrast, the GS researchers achieved considerable growth in the number of AI-related articles published between the two periods (growth of 483%: 6 to 35 articles). GN and GS researchers both contributed to big data and machine learning research pertaining to AI. However, for research topics related to data mining, natural language processing, and neural networks, the GN researchers outperformed the GS researchers.

After the co-citation matrix and the keywords of the analyzed articles were obtained, the VOSviewer [37] was used to visualize the co-citation structure of the IS scholarly network for AI research for the period from 2010 to 2020 to establish a global intellectual structure of AI research in the IS domain. Inn line with previous review studies [13, 15, 16], VOSviewer was used in the present study because it provides for both the visualization and clustering of subnetworks and thus meets the purpose of the present study. In addition, the subgroup structures of the scholarly network were performed by the VOS clustering algorithm. The obtained clustering results yielded a comparative understanding of the main foci of GN and GS researchers in the investigated field.

## Global scholarly network on AI research in IS domain (RQ3)

The SNA was conducted to visualize the co-citation matrix into a network diagram. On the basis of the clustering results, the intellectual structures of subnetworks (i.e., GN and GS networks) were further interpreted and compared. Through the co-citation network ananlysis, we profiled the developmental overview of the IS scholarly network on AI research. Mapping the

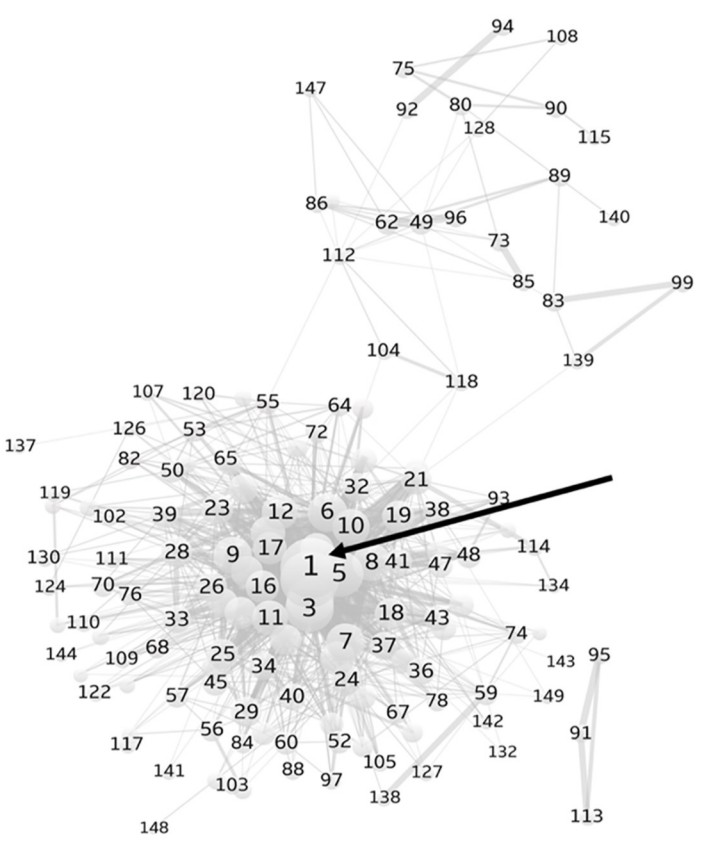

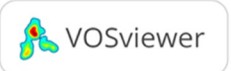

**Fig 2. Full network diagram of information science scholarly network on AI research.**

entire relational structure onto a networking diagram revealed several notable findings relating to multiple perspectives—for example, the main research foci of the GN and GS researchers can be compared. Moreover, all the articles were categorized by author-defined keywords to highlight the main characteristics of AI-related research in the IS domain.

Fig 2 presents a full scholarly network of all 149 articles and the corresponding 8838 citing references, representing the full overview of the contemporary IS scholarly network on AI research from 2010 to 2020. The main interests of the 149 core articles are labeled as nodes with numbers, and each line between two nodes represents a co-citation link indicating the co-citation of two focal articles by a citing article (n = 8838). The thickness of the link represents the weight of the co-citation link. The results suggest that the scholarly network consists of two subnetworks. One loose network is scattered across the upper area of the diagram, whereas the other network, which is dense, is focused in the central area. For example, the most cross-referenced articles (e.g., top co-cited articles [#1–#10], see Table 2) in the central network are indicated by a solid arrow (Fig 2).

## Findings from GN network

The VOS clustering results indicated that all 111 articles written by the GN researchers can be clustered into nine subgroups (Fig 3). The first cluster (N1) consists of 22 articles that are mainly position or review papers. Although several studies have discussed the growth of big

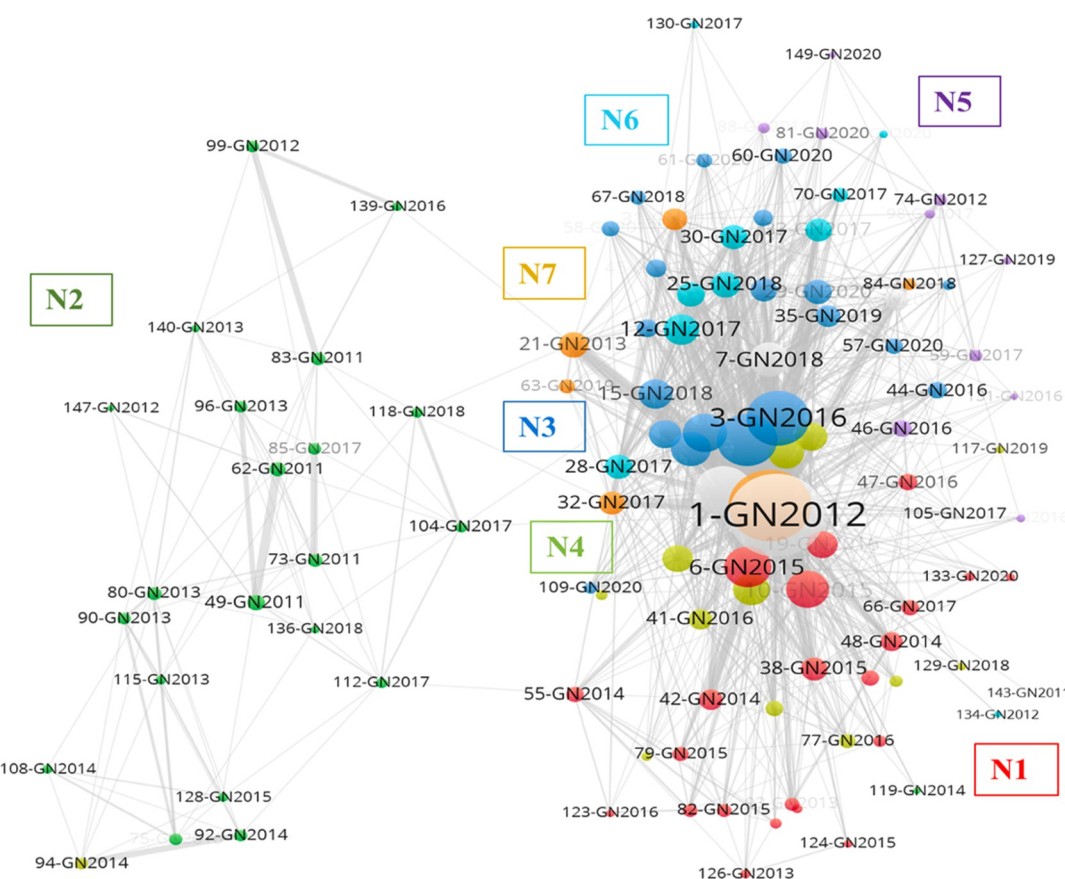

**Fig 3. Global North network perspective of information science scholarly network on AI research.**

data in society and business (#10 and #19) [33, 38], other studies have emphasized concerns pertaining to privacy and ethical judgment (#38, #42, #145, and #87) [39–42]. Other scholars have highlighted the application of big data in various contexts, such as the clinical (#120) [43] and biomedical fields (#55) [44], social media (#79, #126, and #124) [45–47], and mobile applications (#93) [48]. Notably, the main keywords used in this group are "big data" and "big data analytics" (total count of 22); among the top 10 most frequently co-cited articles, two articles in this group (#6 and #10) [32, 33] focused on research on the transformation of businesses and strategies through AI-enhanced technologies.

Consisting of 22 articles, the second cluster (N2) is focused on AI-enhanced clinical research (mainly published in the *JAMIA*). This cluster exhibits two research foci, namely the analysis of data from pathology reports or that from clinical trials and social media texts. The main keywords in this group are "natural language processing," "health records," "machine learning," and "text mining." For the investigation of clinical documents, researchers have attempted to develop machine learning–based, text mining, or natural language processing methods for extracting clinically relevant medical problems, tests, and treatments that improve performance in various situations such as cancer diagnosis and treatment (#90, #75, #80, #83, and #99) [49–53] and influenza detection (#108) [54]. Social media was also revealed to provide an extensive data repository of consumer experiences and information. Several well-designed neural network methods can be used to extract e-cigarette adverse events (#118) [55] and adverse drug reactions (#104) [56].

The third cluster (N3) consists of 20 articles that were published between 2015 and 2020. In this cluster, the main research focus relates to big data, including big data analysis, big data, dynamic functions, and business value. For research content, the main topic concerns the effective use of big data in organizations, with studies focusing on the resource-based view (#3 and #29) [34, 57], firm performance (#11) [58], supply chain management (#4, #15, and #44) [30, 59, 60], business value (#9) [36], automation (#20 and #24) [3, 61], customer relationship management (#58) [62], and human relationships (#61, #67, and #97) [63–65]. Two frequently co-cited articles, namely Gupta and George (#3) [34] and Chen et al. (#4) [30], have verified the benefits of using big data analytics to improve firm performance and create organizational value.

The fourth cluster (N4) contains 13 articles that were published between 2014 and 2019. In this cluster, the major keywords are "big data," "big data analytics," "machine learning," "deep learning," and "data quality." The main topics are related to marketing research, including marketing resource allocation (#41) [66], word of mouth in social media (#129) [67], storytelling (#110) [68], identification of crucial sellers through a text mining system (#102) [69], customer relationships (#26) [70], and privacy concerns about sharing data in large transactional databases (#65) [71].

The fifth cluster (N5) consists of 10 articles that were published between 2012 and 2020. In addition to the term "big data," other keywords include "social network analysis," "text mining," "technology acceptance model (TAM)," and "behavioral intention." On the basis of keywords and article content, we determined that the articles in N5 were related to the acceptance of big data analytical tools, such as the modeling of users' adoption of big data (#46) [72], mobile commerce (#59) [73], and data mining tools (#81 and #74) [74, 75]. Among them, the TAM is still the most widely adopted theory (#59, #81, and #74) [73–75]. For analyzing big data, most researchers in this subgroup adopted social network and neural network methods (#59, #98, #127, and #149) [73, 76–78] as the data/text mining tools.

The sixth cluster (N6) consists of nine articles, most of which were published after 2017. The main keywords include "big data," "knowledge management," "business analytics," and "business intelligence." We can describe this cluster as a knowledge management–themed cluster. The related research topics include research review (#12) [79], large-scale cognitive data (#70, #27, and #33) [7, 80, 81], knowledge management systems (#28) [82], personal knowledge management (#130) [83], and human resources with big data professions (#25) [84]. The last cluster (N7) contains six articles published between 2013 and 2019. The main keywords are "big data," "predictive analytics," and "tourism." In addition to a review study on customer relationship management, the other studies in this cluster were related to hospitality research and covered topics such as tourism behavior (#32) [85], smart tourism designation (#31) [86], social media in the hotel industry (#63) [87], and the pizza industry (#21) [88]. Notably, the subgroups in the final two clusters all contained fewer than five articles and were not characterized and named. The seven clusters of the GN network accounted for 92% of the GN research articles that were included in the present study. Table 4 summarizes the name of each cluster, the time period during which articles were published, and the author-defined keywords of each article.

## Findings from GS network

Fig 4 illustrates the GS researchers' scholarly network on AI research. The VOS clustering results indicated that all 39 articles published by the GS researchers can be clustered into eight subgroups. The first emerging group (S1) consists of the seven latest articles that were published between 2017 and 2019. Their main research topics are machine learning and big data

**Table 4. Summary of research foci in Global North network.**

| Cluster (articles) | Name of sub-network | Active years | Main research foci (number of articles) |
|---|---|---|---|
| N1 (22) | Research review | 2013–2020 | • big data (15)<br>• privacy (3)<br>• big data analytics (2)<br>• social network analysis (2)<br>• social data (2) |
| N2 (22) | Clinical research | 2011~2018 | • natural language processing (9)<br>• health records (8)<br>• machine learning (6)<br>• text mining (4)<br>• extraction (3)<br>• recurrent neural network (2) |
| N3 (20) | Organization, Customer relationships management | 2015~2020 | • big data analytics (11)<br>• big data (7)<br>• dynamic capabilities (4)<br>• artificial intelligence (3)<br>• business value (3)<br>• machine learning (2)<br>• firm performance (2)<br>• resource-based view (2) |
| N4 (13) | Marketing | 2014~2019 | • big data (6)<br>• big data analytics (2)<br>• machine learning (2)<br>• deep learning (2)<br>• data quality (2) |
| N5 (10) | Acceptance modeling | 2012~2020 | • big data (6)<br>• social network analysis (2)<br>• text mining (2)<br>• TAM (2)<br>• Behavioral intention (2) |
| N6 (9) | Knowledge management | 2012~2020 | • big data (6)<br>• knowledge management (4)<br>• business analytics (3)<br>• business intelligence (2)<br>• Human resource management (2)<br>• information management (2) |
| N7 (6) | Hospitality | 2013~2019 | • big data (3)<br>• predictive analytics (2)<br>• tourism (2) |

analytics. Several articles in this category were position or review papers on big data analytics (#34) [89] and big data adoption (#54) [8], and other studies investigated the adoption of big data among managers (#14 and #40) [90, 91]. The second cluster (S2) of GS research consists of seven articles that were published between 2014 and 2019. The keywords in S2 are "big data," "knowledge," "oil and gas," and "co-creation." The articles in S2 addressed the value creation of applying big data in various contexts, such as in the fashion industry (#18) [48], the oil and gas industry (#39 and #106) [92, 93], health-care services (#51) [94], taxi services (#111) [95], stock markets (#76) [96], and big data in smart cities (#22) [97]. Table 5 summarizes the research foci of the GS network.

## Conclusion and limitations

The present study revealed the IS scholarly network on AI research from 2010 to 2020. On the basis of systematic searches of and data retrieved from the WoS, 149 core articles and a corresponding 8838 follow-up citing articles were identified. In accordance with the definition provided by Confraria et al. [19], the identified articles were classified into GN and GS networks.

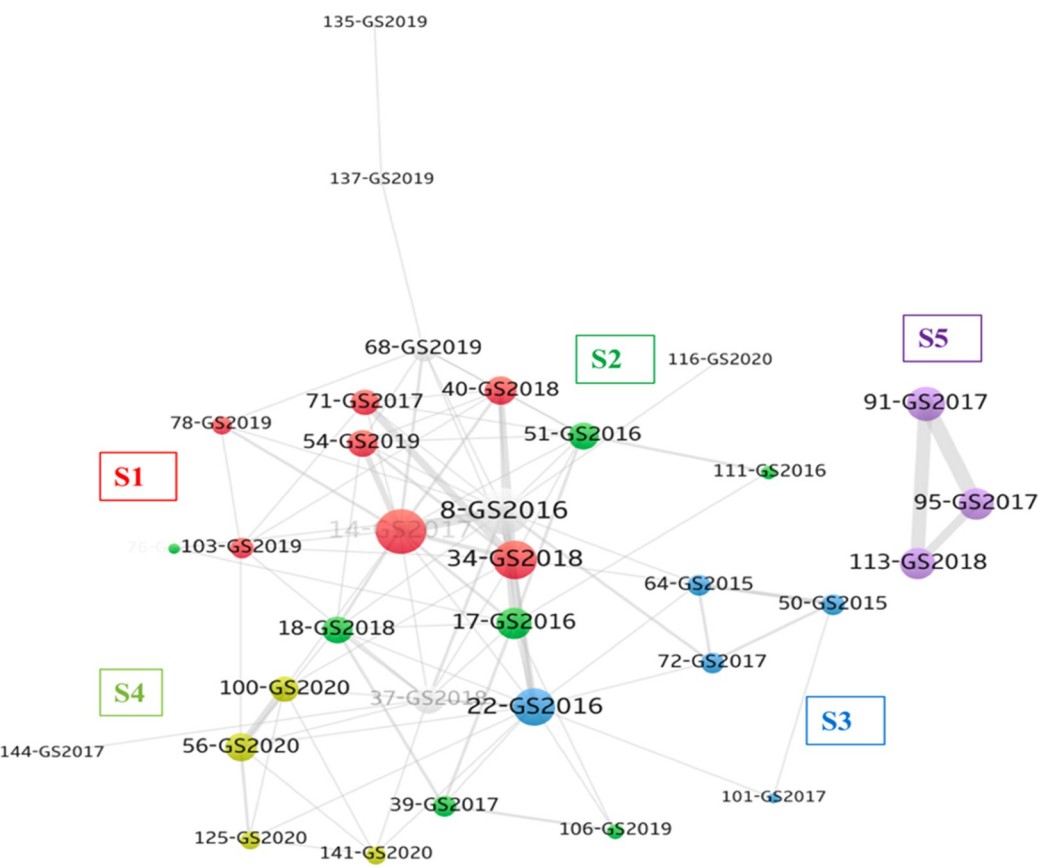

**Fig 4. Global South network perspective on information science scholarly network on AI research.**

A co-citation network analysis was performed to reveal the citation-based networking structure of the literature.

Three key findings regarding the scholarly network of AI research in the IS domain were made. First, in terms of publication patterns, the United States, Australia, and the United

**Table 5. Summary of research foci in Global South network.**

| Cluster (articles) | Name of sub-network | Active years | Main research foci (number of articles) |
|---|---|---|---|
| S1 (7) | Adoption literature | 2017~2019 | • machine learning (3)<br>• big data analytics (3)<br>• big data (2)<br>• technology-organization-environment (2) |
| S2 (7) | Big data applications | 2014~2019 | • big data (4)<br>• knowledge (4)<br>• oil and gas (2)<br>• co-creation (2) |
| S3 (5) | Review | 2012~2017 | • big data (4)<br>• internet of things (2)<br>• big data research (2) |
| S4 (4) | Innovation and performance | 2020 | • big data (4)<br>• innovation performance (2)<br>• high-tech enterprises (1) |
| S5 (3) | Geospatial big data | 2017–2019 | • geospatial big data (2)<br>• social sensing (1)<br>• population mapping (1) |

Kingdom were the most productive GN countries with respect to AI research in the IS domain, and China and India were the most influential countries among the GS countries in the investigated field. Next, the top 10 most frequently co-cited AI research articles in the IS domain were identified. Although 9 of the top 10 articles were contributed by GN researchers, one frequently co-cited article from the GS was also identified. The global scholarly network of AI research in the IS domain and the two subnetworks in the GN and GS were visualized using co-citation network analysis. The results of the scholarly network provide a geographic understanding of the global development of AI research.

A comparison of the applications of AI research in the GN and GS networks revealed several findings. First, most AI research in the GN network focused on AI applications in the areas of organization, customer relationship management (N3), marketing (N4), knowledge management (N6), and hospitality (N7). In particular, the latest trends in AI applications relate to business performance and organizational value creation. This finding is meaningful because most GN countries (e.g., USA and Japan) are known for their service industries. Therefore, AI research and applications in marketing and hospitality are expected to continue growing in the future. By contrast, most GS researchers focused on AI-related big data applications. We also discovered that the latest trend in the GS network relates to innovative technology and its outcomes, such as GIS big data analysis (S5) and performance research (S4), which are expected to dominate future AI research trends. Finally, a research focus shared in common by GN and GS researchers was technology adoption research, which includes topics such as AI-related products and services.

Several changes in research trends between the two time periods (2010–2015 and 2016–2020) are evident. First, GN researchers conducted research in the AI field earlier and published more research articles than GS researchers did during the first research period (2010–2015). During this period, GN scholars laid a solid foundation for AI research concerning its application to traditional business functions (e.g., supply chain management, knowledge management, marketing, and decision support systems) and established several notable theories (e.g., the TAM and resource-based theory [RBT]). During the second investigated period (2016–2020), the number of articles published by GS countries (e.g., China and India, whose economies were growing rapidly) increased rapidly, and GS scholars focused on specific research topics (e.g., innovation and performance). On the basis of the aforementioned results, several suggestions are proposed for future research in terms of research themes, contexts, and theories; business activities; and types of research (Fig 5).

First, among the research themes, big data (big data analytics) accounted for the largest proportion of AI research, followed by machine learning, natural language processing, and text mining. This finding echoes the suggestions of field researchers that the capacity for handling big data (e.g., big data management [6], data richness [7], and bag data adoption [8]) has become a competitive advantage for today's businesses and companies. Second, with respect to research context, the development of technology applications has increased the role of social media not only in the daily lives of individuals but also in research, especially in terms of the ability to collect tremendous volumes of consumer data. Therefore, clinical records, privacy problems, and financial or banking services are the key research topics for future AI research. Third, in terms of business activities, marketing-related activities accounted for the largest portion of AI research, followed by decision support systems and supply chain management. In future studies, researchers should develop AI-based applications for managing marketing intelligence and supporting decision-making in various business activities. Finally, in terms of supported theories, the TAM and RBT are the most widely used theoretical frameworks. The TAM can be applied for research-based empirical surveys, especially concerning the acceptance of big data analytical tools, whereas the RBT can be applied to improve the

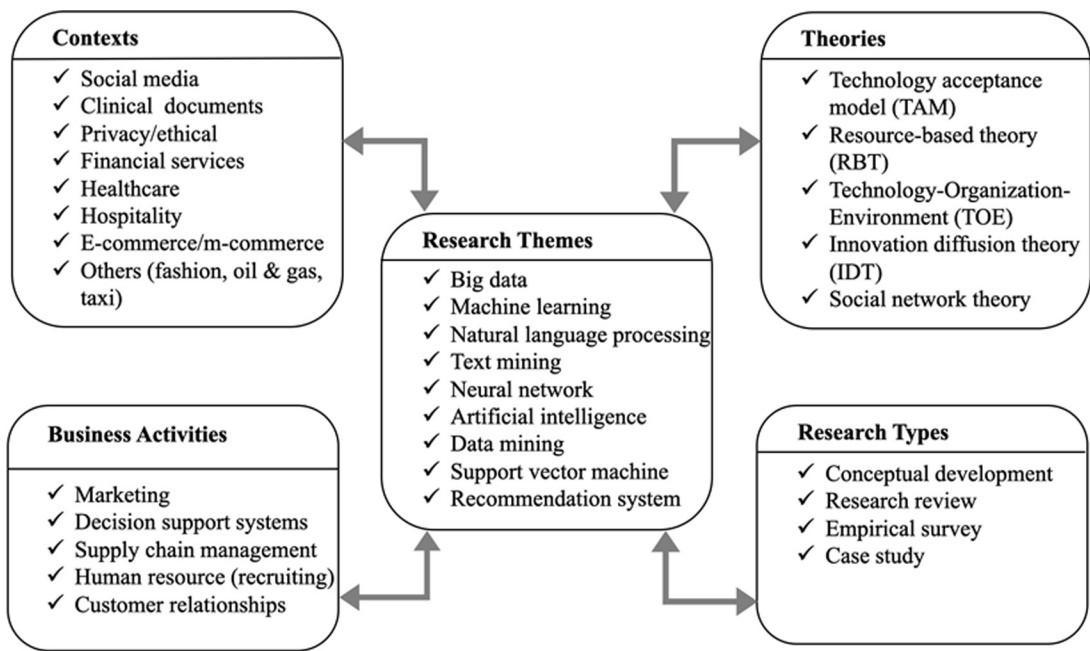

**Fig 5. Proposed framework for future studies on AI research in information science domain.**

competitiveness and performance of a company. For future research, the aforementioned theories can serve as research frameworks that provide further theoretical support for empirical surveys and case studies in AI research. Overall, the proposed framework can provide a foundation and guidance for future researchers who are interested in AI research.

Several suggestions pertaining to the methodology of future studies can be provided. First, for data sets, we only collected articles from the WoS, which is regarded as a database with a highly rigorous collection of articles. To expand the sources of research articles, future research should collect articles from other reputed databases (e.g., Scopus and IEEE Xplore). Furthermore, for the author co-citation analysis of the literature, the present study used the affiliations of first authors as the basis for analysis. Future researchers should conduct further analysis using the full list of authors of investigated articles as the basis for categorizing GN and GS networks in other research fields. Finally, for research scope, the present study mainly focused on the IS domain. Future studies should consider the development of more AI-related applications for other fields such as deep learning architecture and algorithm development.

## Supporting information

**S1 Checklist. PRISMA 2009 checklist.**
(DOC)

**S1 Appendix. The query of the search.**
(DOCX)

## Author Contributions

**Conceptualization:** Kai-Yu Tang, Chun-Hua Hsiao.

**Data curation:** Kai-Yu Tang.

**Formal analysis:** Kai-Yu Tang, Chun-Hua Hsiao.

**Funding acquisition:** Kai-Yu Tang, Chun-Hua Hsiao, Gwo-Jen Hwang.

**Investigation:** Gwo-Jen Hwang.

**Methodology:** Kai-Yu Tang.

**Project administration:** Kai-Yu Tang, Chun-Hua Hsiao, Gwo-Jen Hwang.

**Resources:** Kai-Yu Tang, Chun-Hua Hsiao, Gwo-Jen Hwang.

**Supervision:** Kai-Yu Tang, Chun-Hua Hsiao, Gwo-Jen Hwang.

**Validation:** Kai-Yu Tang, Chun-Hua Hsiao, Gwo-Jen Hwang.

**Visualization:** Kai-Yu Tang.

**Writing – original draft:** Kai-Yu Tang, Chun-Hua Hsiao.

**Writing – review & editing:** Chun-Hua Hsiao.

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
