## [Decision Letter · Decision Letter 0]

29 Dec 2021

PONE-D-21-34311A systematic review of scholarly network on artificial intelligence research: global north and global south perspectivesPLOS ONE

Dear Dr. Tang,

Thank you for submitting your manuscript to PLOS ONE. After careful consideration, we feel that it has merit but does not fully meet PLOS ONE’s publication criteria as it currently stands. Therefore, we invite you to submit a revised version of the manuscript that addresses the points raised during the review process.

We look forward to receiving your revised manuscript.

Kind regards,

Fu Lee Wang

Academic Editor

PLOS ONE

Journal Requirements:

Reviewers' comments:

Reviewer's Responses to Questions

**Comments to the Author**

1. Is the manuscript technically sound, and do the data support the conclusions?

Reviewer #1: Yes

Reviewer #2: Yes

2. Has the statistical analysis been performed appropriately and rigorously? 

Reviewer #1: Yes

Reviewer #2: Yes

3. Have the authors made all data underlying the findings in their manuscript fully available?

Reviewer #1: Yes

Reviewer #2: Yes

4. Is the manuscript presented in an intelligible fashion and written in standard English?

Reviewer #1: Yes

Reviewer #2: Yes

5. Review Comments to the Author

Reviewer #1: This study presents a geographical understanding of scholarly network on artificial intelligence. Specifically, it focuses on publication pattern and the most productive countries, main applications of AI research and research foci, core articles, intellectual structure, and scholarly networks on AI research. Findings can be helpful to provide a comprehensive overview of AI research. Before it can be accepted for publication, there are minor issues to be addressed.

First, justify the use of 2010-2015 and 2016-2020. Why divide the time span this way.

Second, add more AI-related review papers, especially those based on bibliometric analysis:

Chen, X., Tao, X., Wang, F. L., & Xie, H. (2021). Global research on artificial intelligence-enhanced human electroencephalogram analysis. Neural Computing and Applications, 1-39.

Chen, X., Chen, J., Cheng, G., & Gong, T. (2020). Topics and trends in artificial intelligence assisted human brain research. PloS one, 15(4), e0231192.

Also, some papers in this journal that adopts similar social network and framework proposal strategies, for example, “Affective states in digital game-based learning: Thematic evolution and social network analysis”

Third, more explanation and discussion on the proposed framework can be added. Also, based on your findings, what specific suggestions for future research on AI can be provided.

Furthermore, highlight more about the significance of this study in conclusion section.

Finally, double-check both definition and usage of acronyms: every acronym should be defined only once (at the first occurrence) and always used afterwards (except for the abstract). Following this idea, “document co-citation analysis” in page 6 should be DCA. Also check the other acronyms.

Reviewer #2: This paper presents a systematic review of scholarly networks on artificial intelligence research from 2010~2020 from the bibliometric global north and global south perspectives. The paper is well-written and provides insights into the geological traits of researchers. Nevertheless, there are several issues that can be further addressed to improve the paper's quality.

1. There has been a large body of bibliometric surveys in the AI area focusing on a wide spectrum of topics. The authors are highly suggested to justify the motivation of addressing the perspective of the global north and global south, which is relatively coarse-grained categorization, especially in a globalization narrative. The conclusion made based on such a categorization seems not to provide any insight into the future development of AI. The authors may need to discuss how does the conclusion relate to GN and GS.

2. And I am also curious if a paper is considered as GN or GS if it is a collaboration between institutions from both categories. I tried to find the answer in the manuscript, please ignore if I miss something.

3. Similar to 1, it is also better if the reason of dividing last decade into 2 five-year parts can be given. Although I can see that the number of publications and their geographical distribution is distinctive in these two parts, it is still necessary to discuss the change from the GN and GS perspective. Otherwise, it is less meaningful to highlight the GN and GS in this work. Please justify.

4. Another major concern from my side is the credibility and coverage of selected publications. The paper bearing the title of "systematic review" of AI, which nowadays is an incredibly large area with a lot of subfields. Therefore, it is natural to ask how representative the reviewed papers are in the overall AI domain. However, the paper seems to focus more on information science and some domain applications rather than the AI & deep learning architecture and algorithm development. In my understanding, although AI is derived from IS and some techniques are deeply rooted in IS, current AI research is much more than an independent discipline. Maybe the authors need to further clarify their research scope. The manuscript in the current form is not fully qualified as a "systematic review".

6. PLOS authors have the option to publish the peer review history of their article (what does this mean?). If published, this will include your full peer review and any attached files.

Reviewer #1: No

Reviewer #2: No

---

## [Author Response · Author response to Decision Letter 0]

1 Mar 2022

Reviewer #1: This study presents a geographical understanding of scholarly network on artificial intelligence. Specifically, it focuses on publication pattern and the most productive countries, main applications of AI research and research foci, core articles, intellectual structure, and scholarly networks on AI research. Findings can be helpful to provide a comprehensive overview of AI research. Before it can be accepted for publication, there are minor issues to be addressed.

First, justify the use of 2010-2015 and 2016-2020. Why divide the time span this way.

Responses and Revisions:

Thank you for your question. The main reason for using 2010-2015 and 2016-2020 is to compare the development of AI research in the last decade, providing research evidence to identify the direction for future research. Moreover, some similar settings can be found in earlier review studies of the field. For example, Hwang & Tu (2021) conducted a three-stage (1996-2010; 2011-2015; 2016-2020) survey to analyze the trend of AI in mathematics education. Chen et al. (2020) compared the AI-assisted brain research articles published between two periods (2010-2014; 2015-2018).

To address this issue, we have provided a new paragraph to justify the setting of the research period used in this paper. The revision is as follows.

Review studies have also reported that comparing different research periods can help researchers to identify potential trends in a research field. For instance, Sousa et al. [9] reviewed AI studies that were conducted in the public sector between 2000 to 2018 and compared each 3-year period within this timeframe to identify changes in publication patterns. Chen et al. [10] compared the AI-assisted brain research published between the years 2010 and 2014 and between the years 2015 and 2018. Hwang and Tu [13] conducted a three-stage (1996–2010, 2011–2015, 2016–2020) survey to analyze AI trends in mathematics education.

Second, add more AI-related review papers, especially those based on bibliometric analysis:

Chen, X., Tao, X., Wang, F. L., & Xie, H. (2021a). Global research on artificial intelligence-enhanced human electroencephalogram analysis. Neural Computing and Applications, 1-39.

Chen, X., Chen, J., Cheng, G., & Gong, T. (2020). Topics and trends in artificial intelligence assisted human brain research. PloS one, 15(4), e0231192.

Also, some papers in this journal that adopts similar social network and framework proposal strategies, for example, “Affective states in digital game-based learning: Thematic evolution and social network analysis”

Responses and Revisions:

Thanks for the advice. We have added the suggested literature in an additional paragraph as follow:

Scholars have also conducted bibliometric analyses to examine the knowledge structure of AI-related applications in various research areas. For example, de Sousa et al. [9] conducted a systematic literature review to identify research topics and trends relating to AI in the public sector. Chen et al. [10,11] performed a bibliometric analysis to investigate AI-related applications in the medical field, such as an AI-enhanced electroencephalogram method for conducting human brain research. Other educators have combined bibliometric and content analysis to explore mainstream AI-assisted education research [12,13].

New added references are as follows.

9. de Sousa, W. G., de Melo, E. R. P., Bermejo, P. H. D. S., Farias, R. A. S., & Gomes, A. O. How and where is artificial intelligence in the public sector going? A literature review and research agenda. Government Information Quarterly. 2019; 36(4): 101392.

10. Chen, X., Chen, J., Cheng, G., & Gong, T. Topics and trends in artificial intelligence assisted human brain research. PLoS ONE. 2020; 15(4): e0231192.

11. Chen, X., Tao, X., Wang, F. L., & Xie, H. Global research on artificial intelligence-enhanced human electroencephalogram analysis; Neural Computing and Applications. (2021a): https://doi.org/10.1007/s00521-020-05588-x

12. Chen, X., Zou, D., Kohnke, L., Xie, H., & Cheng, G. Affective states in digital game-based learning: Thematic evolution and social network analysis. PLoS ONE. 2021b; 16(7): e0255184.

13. Hwang, G. J., & Tu, Y. F. Roles and research trends of artificial intelligence in mathematics education: A bibliometric mapping analysis and systematic review. Mathematics. 2021; 9(6): 584.

Third, more explanation and discussion on the proposed framework can be added. Also, based on your findings, what specific suggestions for future research on AI can be provided.

Responses and Revisions:

Thank you. According to the reviewer’s comment, we have added more explanation and discussion on the proposed framework. Some suggestions for future research were also provided in this current version. The revision is as follows.

Several changes in research trends between the two time periods (2010–2015 and 2016–2020) are evident. First, GN researchers conducted research in the AI field earlier and published more research articles than GS researchers did during the first research period (2010–2015). During this period, GN scholars laid a solid foundation for AI research concerning its application to traditional business functions (e.g., supply chain management, knowledge management, marketing, and decision support systems) and established several notable theories (e.g., the TAM and resource-based theory [RBT]). During the second investigated period (2016–2020), the number of articles published by GS countries (e.g., China and India, whose economies were growing rapidly) increased rapidly, and GS scholars focused on specific research topics (e.g., innovation and performance). On the basis of the aforementioned results, several suggestions are proposed for future research in terms of research themes, contexts, and theories; business activities; and types of research (Figure 5).

Fig. 5. Proposed framework for future studies on AI research in information science domain

First, among the research themes, big data (big data analytics) accounted for the largest proportion of AI research, followed by machine learning, natural language processing, and text mining. This finding echoes the suggestions of field researchers that the capacity for handling big data (e.g., big data management [6], data richness [7], and bag data adoption [8]) has become a competitive advantage for today’s businesses and companies. Second, with respect to research context, the development of technology applications has increased the role of social media not only in the daily lives of individuals but also in research, especially in terms of the ability to collect tremendous volumes of consumer data. Therefore, clinical records, privacy problems, and financial or banking services are the key research topics for future AI research. Third, in terms of business activities, marketing-related activities accounted for the largest portion of AI research, followed by decision support systems and supply chain management. In future studies, researchers should develop AI-based applications for managing marketing intelligence and supporting decision-making in various business activities. Finally, in terms of supported theories, the TAM and RBT are the most widely used theoretical frameworks. The TAM can be applied for research-based empirical surveys, especially concerning the acceptance of big data analytical tools, whereas the RBT can be applied to improve the competitiveness and performance of a company. For future research, the aforementioned theories can serve as research frameworks that provide further theoretical support for empirical surveys and case studies in AI research. Overall, the proposed framework can provide a foundation and guidance for future researchers who are interested in AI research.

Furthermore, highlight more about the significance of this study in conclusion section.

Responses and Revisions:

Thank you. We have revised the conclusion section to highlight the significance of this current study. The revision is as follows.

The present study revealed the IS scholarly network on AI research from 2010 to 2020. On the basis of systematic searches of and data retrieved from the WoS, 149 core articles and a corresponding 8838 follow-up citing articles were identified. In accordance with the definition provided by Confraria et al. [19], the identified articles were classified into GN and GS networks. A co-citation network analysis was performed to reveal the citation-based networking structure of the literature. 

Three key findings regarding the scholarly network of AI research in the IS domain were made. First, in terms of publication patterns, the United States, Australia, and the United Kingdom were the most productive GN countries with respect to AI research in the IS domain, and China and India were the most influential countries among the GS countries in the investigated field. Next, the top 10 most frequently co-cited AI research articles in the IS domain were identified. Although 9 of the top 10 articles were contributed by GN researchers, one frequently co-cited article from the GS was also identified. The global scholarly network of AI research in the IS domain and the two subnetworks in the GN and GS were visualized using co-citation network analysis. The results of the scholarly network provide a geographic understanding of the global development of AI research.

A comparison of the applications of AI research in the GN and GS networks revealed several findings. First, most AI research in the GN network focused on AI applications in the areas of organization, customer relationship management (N3), marketing (N4), knowledge management (N6), and hospitality (N7). In particular, the latest trends in AI applications relate to business performance and organizational value creation. This finding is meaningful because most GN countries (e.g., USA and Japan) are known for their service industries. Therefore, AI research and applications in marketing and hospitality are expected to continue growing in the future. By contrast, most GS researchers focused on AI-related big data applications. We also discovered that the latest trend in the GS network relates to innovative technology and its outcomes, such as GIS big data analysis (S5) and performance research (S4), which are expected to dominate future AI research trends. Finally, a research focus shared in common by GN and GS researchers was technology adoption research, which includes topics such as AI-related products and services.

Finally, double-check both definition and usage of acronyms: every acronym should be defined only once (at the first occurrence) and always used afterwards (except for the abstract). Following this idea, “document co-citation analysis” in page 6 should be DCA. Also check the other acronyms.

Responses and Revisions:

Thank you for the reminder. We have carefully checked all abbreviations used in this study. 

Reviewer #2: This paper presents a systematic review of scholarly networks on artificial intelligence research from 2010~2020 from the bibliometric global north and global south perspectives. The paper is well-written and provides insights into the geological traits of researchers. Nevertheless, there are several issues that can be further addressed to improve the paper's quality.

1. There has been a large body of bibliometric surveys in the AI area focusing on a wide spectrum of topics. The authors are highly suggested to justify the motivation of addressing the perspective of the global north and global south, which is relatively coarse-grained categorization, especially in a globalization narrative. The conclusion made based on such a categorization seems not to provide any insight into the future development of AI. The authors may need to discuss how does the conclusion relate to GN and GS.

Responses and Revisions:

Thank you for your suggestions. We have revised the manuscript to justify the motivation of this current study. We have also provided more discussions for the future development of AI. The revision is as follows.

(Introduction)

For global development research, North–South divisions are often based on the political and socioeconomic dimensions of the literature (e.g., Martin, 1988). However, on the basis of the high growth rate of a “scientific culture” in several countries [19], such as China and India, several researchers have suggested that scientific development can be explored from the perspective of the Global North (GN) and Global South (GS) [19]. This typology meets the primary purpose of the present study, which is to explore global AI-related research developments. Therefore, in accordance with the research conducted by Confraria et al. [19], we defined the GN to include North America, Western Europe, and the developed regions of East Asia and defined the GS to include Africa, Latin America, developing regions of Asia, and the Middle East.

(Conclusion)

A comparison of the applications of AI research in the GN and GS networks revealed several findings. First, most AI research in the GN network focused on AI applications in the areas of organization, customer relationship management (N3), marketing (N4), knowledge management (N6), and hospitality (N7). In particular, the latest trends in AI applications relate to business performance and organizational value creation. This finding is meaningful because most GN countries (e.g., USA and Japan) are known for their service industries. Therefore, AI research and applications in marketing and hospitality are expected to continue growing in the future. By contrast, most GS researchers focused on AI-related big data applications. We also discovered that the latest trend in the GS network relates to innovative technology and its outcomes, such as GIS big data analysis (S5) and performance research (S4), which are expected to dominate future AI research trends. Finally, a research focus shared in common by GN and GS researchers was technology adoption research, which includes topics such as AI-related products and services. 

Several changes in research trends between the two time periods (2010–2015 and 2016–2020) are evident. First, GN researchers conducted research in the AI field earlier and published more research articles than GS researchers did during the first research period (2010–2015). During this period, GN scholars laid a solid foundation for AI research concerning its application to traditional business functions (e.g., supply chain management, knowledge management, marketing, and decision support systems) and established several notable theories (e.g., the TAM and resource-based theory [RBT]). During the second investigated period (2016–2020), the number of articles published by GS countries (e.g., China and India, whose economies were growing rapidly) increased rapidly, and GS scholars focused on specific research topics (e.g., innovation and performance). On the basis of the aforementioned results, several suggestions are proposed for future research in terms of research themes, contexts, and theories; business activities; and types of research (Figure 5).

Fig. 5. Proposed framework for future studies on AI research in information science domain

First, among the research themes, big data (big data analytics) accounted for the largest proportion of AI research, followed by machine learning, natural language processing, and text mining. This finding echoes the suggestions of field researchers that the capacity for handling big data (e.g., big data management [6], data richness [7], and bag data adoption [8]) has become a competitive advantage for today’s businesses and companies. Second, with respect to research context, the development of technology applications has increased the role of social media not only in the daily lives of individuals but also in research, especially in terms of the ability to collect tremendous volumes of consumer data. Therefore, clinical records, privacy problems, and financial or banking services are the key research topics for future AI research. Third, in terms of business activities, marketing-related activities accounted for the largest portion of AI research, followed by decision support systems and supply chain management. In future studies, researchers should develop AI-based applications for managing marketing intelligence and supporting decision-making in various business activities. Finally, in terms of supported theories, the TAM and RBT are the most widely used theoretical frameworks. The TAM can be applied for research-based empirical surveys, especially concerning the acceptance of big data analytical tools, whereas the RBT can be applied to improve the competitiveness and performance of a company. For future research, the aforementioned theories can serve as research frameworks that provide further theoretical support for empirical surveys and case studies in AI research. Overall, the proposed framework can provide a foundation and guidance for future researchers who are interested in AI research.

2. And I am also curious if a paper is considered as GN or GS if it is a collaboration between institutions from both categories. I tried to find the answer in the manuscript, please ignore if I miss something.

Responses and Revisions:

Thank you for the question. We have provided more detailed descriptions to deal with this issue. Following previous author co-citation studies, we used the first author’s affiliation as the basis for counting papers’ belonging country. For those authors who have multiple affiliations, we will manually check and code their main affiliation for the subsequent categorization of GN or GS based on Confraria et al.’s (2017) definition. In this present analysis, we found 13 first-authors with multiple affiliations, such as universities in Denmark and England. However, the two countries were both categorized as global north in this case. From the methodological perspective, research suggestion for future study was also provided. The revisions are listed as follows. 

(Methods)

Furthermore, we retrieved bibliometric data on the affiliations of first authors as provided in the included articles. In accordance with other author co-citation studies [18], the affiliations of first authors were used as the basis for determining the affiliated country for each article. To present an overview of global research productivity in the investigated field, the most productive countries were analyzed. The results indicated that all the first authors of the 149 articles were from 31 countries. A further categorization of the GN and GS was conducted based on the definition proposed by Confraria et al. [19].

(Limitation)

…Furthermore, for the author co-citation analysis of the literature, the present study used the affiliations of first authors as the basis for analysis. Future researchers should conduct further analysis using the full list of authors of investigated articles as the basis for categorizing GN and GS networks in other research fields.

3. Similar to 1, it is also better if the reason of dividing last decade into 2 five-year parts can be given. Although I can see that the number of publications and their geographical distribution is distinctive in these two parts, it is still necessary to discuss the change from the GN and GS perspective. Otherwise, it is less meaningful to highlight the GN and GS in this work. Please justify.

Responses and Revisions:

Thank you for the suggestion. The main reason for using 2010-2015 and 2016-2020 is to compare the development of AI research in the last decade, providing research evidence to identify the direction for future research. Moreover, some similar settings can be found in earlier review studies of the field. For example, Hwang & Tu (2021) conducted a three-stage (1996-2010; 2011-2015; 2016-2020) survey to analyze the trend of AI in mathematics education. Chen et al. (2020) compared the AI-assisted brain research articles published between the two periods (2010-2014; 2015-2018).

We have discussed some changes from the GN and GS perspectives in Results section as follows.

(Results)

On the basis of the aforementioned results, several changes in publication patterns relating to AI research in the IS domain were identified. First, 47 articles were published in the earlier investigated period (2010–2015), and the number of published articles increased to 102 during the subsequent five years (2016–2020). This result revealed an increasing trend in terms of the number of AI-related articles published globally in recent years. Specifically, during the earlier investigated period, the 41 published articles (87%) were mainly written by GN researchers, and only six articles (13%) were published by GS researchers. In the subsequent period (2016–2020), GN researchers published 70 articles, representing 73% of the increase in the number of articles published in the investigated field. However, GS researchers published 32 articles in the subsequent period, which was equivalent to a considerable increase of 433% relative to the earlier investigated years (2010–2015). From the GN and GS perspective, the change in publication patterns indicates the increasing presence of GS researchers in AI research; this finding echoes that of Confraria et al. [19], who reported the high growth of “scientific culture” in several GS countries.

[Table 1. Top 20 most productive countries in terms of AI research in information science domain]

Table 2 lists the highly co-cited articles on AI research in the IS domain, and it reveals that 60% (6 of 10) of such articles were published during the earlier investigated period (2010–2015); the 60% comprised Chen et al. (#1) [27], Gandomi and Haider (#2) [29], Chen et al. (#4) [30], Kwon et al. (#5) [31], Constantiou and Kallinikos (#6) [32], and Loebbecke and Picot (#10) [22]. The other 40% of the highly referenced articles were published in the recent five years (2016–2020); they comprised Gupta and George (#3) [34], Raguseo (#7) [35], Rehman et al. (#8) [28], and Grover et al. (#9) [36]. Of note, these highly referenced articles mainly focused on big data–related research, including big data analytics (n = 7; e.g., Chen et al. [27]), big data technologies [29], and big data capabilities [34]. 

 [Table 2. Co-citation matrix of top 10 cross-referenced articles]

…Table 3 lists the top five keywords and provides information on the number of articles in which each keyword is used. An overall comparison of the frequency of keyword use during the two investigated periods (2010–2015 and 2016–2020) revealed that “big data” was the most studied topic (used by 105 articles), especially during the later five years (increase from 21 articles between 2010 and 2015 to 84 articles between 2016 and 2020). In addition to “big data,” the other top five keywords were “machine learning (20),” “text/data mining (19),” “natural language processing (14),” and “neural network (7).” Although the keywords “machine learning” (increase from 7 to 13) and “neural network” (increase from 1 to 6) exhibited increasing trends in terms of frequency of use, the keywords “natural language processing” (decrease from 12 to 7) and “neural network” (decrease from eight to six) exhibited slight downward trends from the first to the second investigated periods. 

For the top five most researched topics pertaining to AI, both GN and GS researchers collectively published 49 articles in the earlier investigated period (2010–2015), and this number increased to 116 in the second investigated period (2016–2020), exhibiting a 137% overall growth rate. From a global development perspective, the GN researchers achieved a low growth rate (88%) but a considerable increase in the number of published articles between the two periods (from 43 to 82 articles). By contrast, the GS researchers achieved considerable growth in the number of AI-related articles published between the two periods (growth of 483%: 6 to 35 articles). GN and GS researchers both contributed to big data and machine learning research pertaining to AI. However, for research topics related to data mining, natural language processing, and neural networks, the GN researchers outperformed the GS researchers.

[Table 3. Most researched AI-related keywords among Global South and Global North scholars (2010–2020)]

4. Another major concern from my side is the credibility and coverage of selected publications. The paper bearing the title of "systematic review" of AI, which nowadays is an incredibly large area with a lot of subfields. Therefore, it is natural to ask how representative the reviewed papers are in the overall AI domain. However, the paper seems to focus more on information science and some domain applications rather than the AI & deep learning architecture and algorithm development. In my understanding, although AI is derived from IS and some techniques are deeply rooted in IS, current AI research is much more than an independent discipline. Maybe the authors need to further clarify their research scope. The manuscript in the current form is not fully qualified as a "systematic review".

Responses and Revisions:

Thank you. Based on the reviewer’s suggestion, the article title has been changed to “A scholarly network of AI research with an information science focus: Global North and Global South perspectives.” Moreover, we have revised the Introduction section to clarify our research scope. We have also provided some research directions for future study in the section of Conclusion and Limitations. The revisions are as follows.

(Introduction)

The development of the information science (IS) domain through the application of artificial intelligence (AI) has had profound effects in numerous industries (e.g., finance, health care, manufacturing, retail, supply chain, logistics, and utilities). According to the literature, AI is defined as “a system that perceives its environment and takes actions to maximize its ability to achieve its goals” [1,2]. Researchers have also suggested that the development of supercomputing power has made AI the crucial emerging theme in the IS research domain [3].

In addition, IS researchers have highlighted the key role of big data research in the development of AI technology. For example, Surbakti et al. [6] identified factors that influence the effective use of big data to provide guidance for organizations; they identified seven themes that were divided into three groups, namely motivation (perceived organizational benefit), operation (process management, human aspects, systems, and organizational aspects), and supporting mechanisms (e.g., data quality, privacy, and governance).

For global development research, North–South divisions are often based on the political and socioeconomic dimensions of the literature (e.g., Martin, 1988). However, on the basis of the high growth rate of a “scientific culture” in several countries [19], such as China and India, several researchers have suggested that scientific development can be explored from the perspective of the Global North (GN) and Global South (GS) [19]. This typology meets the primary purpose of the present study, which is to explore global AI-related research developments. Therefore, in accordance with the research conducted by Confraria et al. [19], we defined the GN to include North America, Western Europe, and the developed regions of East Asia and defined the GS to include Africa, Latin America, developing regions of Asia, and the Middle East.

In the present study, a co-citation network analysis was conducted to compare the GN and GS with respect to scholarly networks of AI research in the IS domain during the period from 2010 to 2020. Furthermore, to allow for research trend comparisons with the findings of other studies [10–13], the research period for this study was divided into two intervals (i.e., 2010–2015 and 2016–2020). Overall, the present study provides a fresh perspective on the global research dynamics of the scholarly network of AI research in the IS field. The three research questions are specified as follows:

1) Investigation of the publication patterns of AI research in the IS domain (RQ1): From the bibliometric perspective, what are the publication patterns and the most productive countries in the investigated field between 2010 and 2015 and between 2016 and 2020? 

2) Identification of the most frequently cross-referenced AI research articles in the IS domain (RQ2): On the basis of follow-up citations, the present study performed a co-citation analysis to determine the core articles of IS field. The main AI research foci of GN and GS researchers were also analyzed.

3) Visualization of the global scholarly network on AI research in the IS domain (RQ3): From the social networking perspective, what is the intellectual structure of the scholarly IS network on AI research? Through the application of social network analysis and the clustering technique, the citation-based mapping of scholarly networks of AI research in the IS domain was achieved.

(Conclusion and limitations)

…Finally, for research scope, the present study mainly focused on the IS domain. Future studies should consider the development of more AI-related applications for other fields such as deep learning architecture and algorithm development.

---

## [Decision Letter · Decision Letter 1]

23 Mar 2022

A scholarly network of AI research with an information science focus: Global North and Global South perspectives

PONE-D-21-34311R1

Dear Dr. Tang,

We’re pleased to inform you that your manuscript has been judged scientifically suitable for publication and will be formally accepted for publication once it meets all outstanding technical requirements.

Kind regards,

Fu Lee Wang

Academic Editor

PLOS ONE

Additional Editor Comments (optional):

Reviewers' comments:

Reviewer's Responses to Questions

**Comments to the Author**

1. If the authors have adequately addressed your comments raised in a previous round of review and you feel that this manuscript is now acceptable for publication, you may indicate that here to bypass the “Comments to the Author” section, enter your conflict of interest statement in the “Confidential to Editor” section, and submit your "Accept" recommendation.

Reviewer #1: All comments have been addressed

Reviewer #2: All comments have been addressed

2. Is the manuscript technically sound, and do the data support the conclusions?

Reviewer #1: Yes

Reviewer #2: Yes

3. Has the statistical analysis been performed appropriately and rigorously? 

Reviewer #1: Yes

Reviewer #2: Yes

4. Have the authors made all data underlying the findings in their manuscript fully available?

Reviewer #1: Yes

Reviewer #2: Yes

5. Is the manuscript presented in an intelligible fashion and written in standard English?

Reviewer #1: Yes

Reviewer #2: Yes

6. Review Comments to the Author

Reviewer #1: All my previous concerns have been addressed. I feel it can be accepted for publication. Thank you for your hard work on this paper.

Reviewer #2: I want to thank the authors' for addressing my concerns. The current version fits the publication standard. Good luck to your future research.

7. PLOS authors have the option to publish the peer review history of their article (what does this mean?). If published, this will include your full peer review and any attached files.

Reviewer #1: No

Reviewer #2: No

---

## [Editor Report · Acceptance letter]

7 Apr 2022

PONE-D-21-34311R1 

A scholarly network of AI research with an information science focus: Global North and Global South perspectives 

Dear Dr. Tang:

I'm pleased to inform you that your manuscript has been deemed suitable for publication in PLOS ONE. Congratulations! Your manuscript is now with our production department. 

Kind regards, 

on behalf of

Professor Fu Lee Wang 

Academic Editor

PLOS ONE